# Iterative Refinement of Flow Policies in Probability Space for Online Reinforcement Learning

## Abstract

While behavior cloning with flow/diffusion policies excels at learning complex skills from demonstrations, it remains vulnerable to distributional shift, and standard RL methods struggle to fine-tune these models due to their iterative inference process and the limitations of existing workarounds. In this work, we introduce the Stepwise Flow Policy (SWFP) framework, founded on the key insight that discretizing the flow-matching inference process via a fixed-step Euler scheme inherently aligns it with the variational Jordan–Kinderlehrer–Otto (JKO) principle from optimal transport. SWFP decomposes the global flow into a sequence of small, incremental transformations between proximate distributions. Each step corresponds to a JKO update, regularizing policy changes to stay near the previous iterate and ensuring stable online adaptation with entropic regularization. This decomposition yields an efficient algorithm that fine-tunes pre-trained flows via a cascade of small flow blocks, offering significant advantages: simpler/faster training of sub-models, reduced computational/memory costs, and provable stability grounded in Wasserstein trust regions. Comprehensive experiments demonstrate SWFP's enhanced stability, efficiency, and superior adaptation performance across diverse robotic control benchmarks.

## 1 Introduction

Robotic control has rapidly evolved from hand-crafted feedback systems to data-driven, end-to-end learning paradigms. Modern tasks demand not only precise execution of complex behaviors but also robust adaptability to environmental uncertainties. Imitation Learning (IL) has emerged as a dominant approach, with recent advances in Behavior Cloning (BC)—such as diffusion policies Chi et al. (2023); Ke et al. (2024); Ze et al. (2024) and action chunking (Zhao et al., 2023) robots to acquire sophisticated, long-horizon skills from demonstration data. Notably, generative policies based on diffusion or flow matching offer compelling advantages: they natively handle multi-modal action distributions, scale effectively in high-dimensional spaces, and achieve training stability through denoising and score-matching techniques.

Despite these innovations, BC methods remain fundamentally vulnerable to distributional shift—where deviations from training states induce compounding errors (Ross & Bagnell, 2010). Reinforcement Learning (RL) offers a principled path for policy improvement through online interaction (Chen et al., 2024; Wagenmaker et al., 2025; Ren et al., 2025), while learning policies via RL from scratch is often too inefficient for practical use. Unfortunately, fine-tuning expressive generative policies presents unique challenges. Unlike Gaussian policies, diffusion or flow models require iterative inference (often tens of steps), which fundamentally breaks the single-step gradient path required by standard policy-gradient or Q-learning algorithms. Existing workarounds—weighted regression(Lu et al., 2023), implicit reparameterization(Ada et al., 2024), or rejection sampling(Hansen-Estruch et al., 2023)—either introduce large gradient variance or lack a rigorous interpretation in terms of distribution optimization, leaving their stability and convergence properties fundamentally unclear.

In this work, we take steps towards addressing this challenge and seek to understand how we can utilize online interaction to adapt pre-trained flow policies in a simple and efficient way. Our core

insight arises from reinterpreting the fixed-step Euler scheme of flow matching: the multi-step transport process from a noise distribution to an optimized action distribution naturally corresponds to traversing a Wasserstein gradient flow (Santambrogio, 2017; Zhang et al., 2018) in probability space. This perspective allows us to leverage powerful tools from optimal transport. Specifically, we rigorously align this process with the Jordan–Kinderlehrer–Otto (JKO) scheme (Jordan et al., 1998; Mokrov et al., 2021; Xie & Cheng, 2025), a variational principle from optimal transport that provides theoretical guarantees for stable, incremental policy updates. By decomposing the global flow into a sequence of incremental transformations, each JKO step regularizes policy updates to stay near the previous iterate, mitigating instability while enabling efficient optimization.

We instantiate this concept in **Stepwise Flow Policy (SWFP)**, a stable algorithm that fine-tunes pretrained flow policies through a cascade of small, specialized flow blocks. Unlike monolithic flow matching, which directly bridges potentially distant noise and target distributions, SWFP structures the transport into localized steps where consecutive distributions are proximate. This decomposition yields three key advantages: (1) each sub-flow model in fixed step scheme is simpler to train and converges faster, (2) computational and memory costs are significantly reduced, and (3) the JKO alignment provides a theoretically grounded mechanism for stable policy improvement with entropic regularization.

- We propose SWFP, the first framework that unifies Flow-Matching policies with the Jordan–Kinderlehrer–Otto (JKO) proximal operator, establishing a principled bridge between generative inference and regularized policy improvement in reinforcement learning.

- Building on the framework, we devise a practical algorithm that finetunes a pre-trained flow policy through a succession of short, easily trained flow blocks. The method retains the theoretical guarantees of Wasserstein trust regions while remaining practical in high-dimensional continuous-action settings.

- Comprehensive experiments demonstrating SWFP's advantages, including enhanced stability, reduced computational overhead, and superior adaptation efficiency, across diverse robotic control benchmarks.

## 2 RELATED WORK

### 2.1 GENERATIVE MODELS FOR DECISION MAKING

Generative models, particularly those based on diffusion (Ho et al., 2020) and flow-matching (Lipman et al., 2022), have become the state-of-the-art for learning complex robot behaviors. In imitation learning, they excel at capturing multi-modal action distributions from expert data, significantly outperforming standard Behavioral Cloning (Chi et al., 2023; Braun et al., 2024). They have also proven highly effective in offline reinforcement learning (He et al., 2023; Wang et al., 2022; Mao et al., 2024), where models like Diffuser (Ajay et al.) use them for planning reward-conditioned trajectories. However, a crucial limitation persists across these applications: their iterative inference structure and large parameter counts make them fundamentally difficult to fine-tune efficiently using online RL (Uehara et al., 2024; Clark et al., 2023). Our work directly addresses this challenge of adapting powerful, pre-trained generative policies to new online information.

### 2.2 RL WITH DIFFUSION AND FLOW MODELS

To bridge the gap between offline pre-training and online adaptation, several fine-tuning strategies have been explored. One line of work treats the multi-step denoising process as an MDP to apply policy gradients (Black et al., 2024; Fan et al., 2024; Ren et al., 2025), but this often leads to high variance and instability. A second, more indirect approach uses learned value functions to guide or filter the output of a frozen generative policy, for instance via advantage-weighted regression (Goo & Niekum, 2022), rejection sampling (Hansen-Estruch et al., 2023), or reformulating the objective as a supervised learning problem with return conditioning (Chen et al., 2021; Janner et al., 2022; Ajay et al.). While practical, these methods do not improve the underlying policy distribution itself. Additionally, Reinflow (Zhang et al., 2025) converts the deterministic flow path into a stochastic process (SDE) by injecting learnable noise to enable standard policy gradient optimization. Flow

Q-learning (Park et al., 2025b) trains an auxiliary one-step policy to guide the flow policy via distillation. In contrast to these prior works that struggle with stability or are indirect, our method, SWFP, introduces a principled framework for performing direct, stable, and incremental updates to the flow policy's distribution. By grounding our algorithm in the Jordan–Kinderlehrer–Otto (JKO) scheme, we provide a theoretically sound mechanism for online flow policy improvement.

## 3 PRELIMINARIES

**Flow matching.** Continuous Normalizing Flows (CNFs) (Chen et al., 2018) considers the dynamics of the probability density function by probability density path $p : [0,1] \times \mathbb{R}^d \to \mathbb{R}_{\geq 0}$ which transmits between the data distribution $p_1$ and the initial distribution (e.g., Gaussian distribution) $p_0$. The flow $\phi : [0,1] \times \mathbb{R}^d \to \mathbb{R}^d$ is constructed by a vector field $\boldsymbol{v} : [0,1] \times \mathbb{R}^d \to \mathbb{R}^d$ describing the velocity of the particle at position $\boldsymbol{x}$, i.e., the ODE

$$\frac{\mathrm{d}}{\mathrm{d}t}\phi_t(\boldsymbol{x}) = \boldsymbol{v}_t(\phi_t(\boldsymbol{x})), \tag{1}$$

where $\phi_0(\boldsymbol{x}) = \boldsymbol{x}$. In order to ensure that the vector field $\boldsymbol{v}$ generates the probability density path $p_t$, the following *continuity equation* (CE) (Villani, 2013) is required:

$$\frac{\mathrm{d}}{\mathrm{d}t}p_t(\boldsymbol{x}) + \mathrm{div} \cdot [p_t(\boldsymbol{x})\boldsymbol{v}_t(\boldsymbol{x})] = 0, \quad \forall \boldsymbol{x} \in \mathbb{R}^d \tag{2}$$

Given such a process, flow matching models learn a neural network $\boldsymbol{v}_t^\theta$ to learn the ground truth vector field $\boldsymbol{u}_t$ by minimizing their differences, i.e., $\mathcal{L}_{FM}(\theta) = \mathbb{E}_{t,p_t(\boldsymbol{x})}\|\boldsymbol{v}_t^\theta(\boldsymbol{x}) - \boldsymbol{u}_t(\boldsymbol{x})\|_2^2$ with respect to the network parameter $\theta$.

However, the original FM objective is generally intractable because the time-varying distribution $p_t(\boldsymbol{x})$ and the true vector field $\boldsymbol{u}_t(\boldsymbol{x})$ are often intractable. To address this, Conditional Flow Matching (CFM) has been proposed (Lipman et al., 2023), which simplifies the task by conditioning the flow on target samples to derive a tractable objective. The CFM loss is defined as follows:

$$\mathcal{L}_{\mathrm{CFM}}(\theta) = \mathbb{E}_{\substack{t\sim\mathcal{U}(0,1),\\ \boldsymbol{x}_1\sim q(\boldsymbol{x}_1),\\ \boldsymbol{x}\sim p_t(\boldsymbol{x}|\boldsymbol{x}_1)}} \left[\|\boldsymbol{v}_\theta(t,\boldsymbol{x}) - \boldsymbol{u}_t(\boldsymbol{x}|\boldsymbol{x}_1)\|\right]^2 \tag{3}$$

wherein $\boldsymbol{u}_t(\boldsymbol{x}|\boldsymbol{x}_1)$ becomes tractable by defining explicit conditional probability paths from $\boldsymbol{x}_0$ to $\boldsymbol{x}_1$, such as OT-paths or linear interpolationv (in this paper) paths.

**Reinforcement Learning** In this paper, we focus on policy learning in continuous action spaces. We consider a Markov Decision Process (MDP) defined by the tuple $(\mathcal{S}, \mathcal{A}, \mathcal{P}, r, \rho_0, \gamma)$, where $\mathcal{S}$ represents the state space, $\mathcal{A}$ is the continuous action space, $\mathcal{P} : \mathcal{S} \times \mathcal{S} \times \mathcal{A} \to [0, +\infty]$ is the probability density function of the next state $\boldsymbol{s}' \in \mathcal{S}$ given the current state $\boldsymbol{s} \in \mathcal{S}$ and the action $\boldsymbol{a} \in \mathcal{A}$, $r : \mathcal{S} \times \mathcal{A} \to [r_{\min}, r_{\max}]$ is the bounded reward function.

The standard RL aims to learn a policy that maximizes the expected cumulative reward: $\mathbb{E}_\pi[r(\boldsymbol{a})] = \int_\mathcal{A} r(\boldsymbol{a})\mathrm{d}\pi(\boldsymbol{a})$. Shannon entropy is often added as a regularization term to improve exploration and avoid early convergence to suboptimal policies. This gives us the entropy-regularised reward, which is a free energy functional, named by analogy with a similar quantity in statistical mechanics:

$$J(\pi) = \int_\mathcal{A} r(\boldsymbol{a})\mathrm{d}\pi(\boldsymbol{a}) - \beta \int_\mathcal{A} \log \pi(\boldsymbol{a})\mathrm{d}\pi(\boldsymbol{a}) \tag{4}$$

Equation (4) is often interpreted as a free energy functional by analogy with statistical mechanics, and it underlies several state-of-the-art algorithms such as soft Q-learning, soft Actor-Critic and entropy-regularized policy gradient methods (Haarnoja et al., 2017; Nachum et al., 2017; Haarnoja et al., 2018).

We are interested in the process of policy iteration, that is, finding a sequence of policies $(\pi_n)$ converging towards the optimal policy $\pi^*$. Policy iteration is often implemented using gradient ascent according to

$$\pi_{n+1} = \pi_n + \tau \nabla J(\pi_n)$$

In implicit Euler method, if integrated and interpreted as an $L^2$ regularized iterative problem, it is strictly equivalent to finding a solution to the proximal problem:

$$\pi_{n+1} = \arg\min_{\pi} \frac{\|\pi - \pi_n\|^2}{2\tau} - J(\pi) \tag{5}$$

Rather than just the $L^2$ distance between policies for constraining and regularization, one can envision the more general case of any policy distance $d$:

$$\pi_{n+1} = \arg\min_{\pi} \frac{d^2(\pi, \pi_n)}{2\tau} - J(\pi) \tag{6}$$

## 4 METHOD

In this section, we first identify the difficulties in training of flow-based policy in the context of online RL. To mitigate this, we propose our core contribution, SWFP, an iterative flow using JKO scheme. Finally, we give a practical algorithm with soft critic and parallel block training.

### 4.1 FLOW AS RL POLICY

We begin by formalizing our flow-based policy framework. The core architecture employs flow matching in action space $\mathcal{A}$, parameterized by a state- and time-dependent vector field $\boldsymbol{v}_\theta(t, \boldsymbol{s}, \boldsymbol{a})$. The fundamental behavioral cloning objective is given by:

$$\mathcal{L}_{\text{FM}}(\theta) = \mathbb{E}_{\substack{\boldsymbol{s}, \boldsymbol{a}_1 \sim \mathcal{D}, \\ \boldsymbol{a}_0 \sim \mathcal{N}(0, I_d), \\ t \sim \mathcal{U}(0,1)}} \left[ \|\boldsymbol{v}_\theta(t, \boldsymbol{s}, \boldsymbol{a}_t) - \boldsymbol{u}_t(\boldsymbol{a}_t | \boldsymbol{a}_1)\|_2^2 \right] \tag{7}$$

The state-dependent vector field generates a state-dependent flow $\boldsymbol{\phi}_\theta(t, \boldsymbol{s}, \boldsymbol{a}) : [0, 1] \times \mathcal{S} \times \mathbb{R}^d \to \mathbb{R}^d$. For $\boldsymbol{s} \in \mathcal{S}$ and $\boldsymbol{z} \in \mathbb{R}^d$, $\boldsymbol{\phi}_\theta(1, \boldsymbol{s}, \boldsymbol{z})$ maps the noise $\boldsymbol{z} = \boldsymbol{x}_0$ (sampled from the standard normal distribution $\mathcal{N}(0, I_d)$) to the action $\boldsymbol{a} = \mu_\theta(\boldsymbol{s}, \boldsymbol{z})$ by the ODE. $\mu_\theta(\boldsymbol{s}, \boldsymbol{z})$ is a deterministic function from $\mathcal{S} \times \mathbb{R}^d$ to $\mathcal{A}$, but serves as a stochastic policy $\pi_\theta(\boldsymbol{a} | \boldsymbol{s})$ from $\mathcal{S}$ to $\mathcal{A}$ due to the stochasticity of $\boldsymbol{z}$.

During online RL, optimizing the objective in (4) with a maximum entropy constraint provides us with a framework for training stochastic policies. Speically, we consider a general energy-based policies of the form $\pi(\boldsymbol{a} | \boldsymbol{s}) \propto \exp(-\epsilon(\boldsymbol{s}, \boldsymbol{a})/\alpha)$ that is able to model more complex distributions. Following the soft policy iteration algorithm, the policy is updated to fit the target max-entropy policy

$$\pi_{\text{MaxEnt}}(\boldsymbol{a} | \boldsymbol{s}) \propto \exp(Q^{\pi_{\text{old}}}(\boldsymbol{s}, \boldsymbol{a})) \tag{8}$$

where $Q^{\pi_{\text{old}}}(\boldsymbol{s}, \boldsymbol{a})$ is the converged result of soft Bellman update operator.

However, training flow-based policies in online reinforcement learning remains highly challenging due to two fundamental limitation. Firstly, the flow-matching objective (7) becomes intractable in online RL settings, as it requires samples from the entropy-regularized optimal policy (8)—which are unavailable during iterative policy improvement. Secondly, while one might consider treating the reverse generative process as a policy parameterization and backpropagating gradients through the full sampling trajectory, this approach incurs prohibitive computational and memory overhead from recursive gradient computations.

These limitations motivate our novel training framework presented in subsequent sections, which maintains the expressiveness of flow-based policies while addressing these practical constraints.

### 4.2 ITERATIVE FLOW USING JKO SCHEME

In the continuous-time flow trained, let $\boldsymbol{x}_t$ be the ODE solution trajectory satisfying $\dot{\boldsymbol{x}}_t = \boldsymbol{v}_\theta(t, \boldsymbol{x}_t)$ on $[0, T]$, the flow mapping can be written as:

$$F_\theta(\boldsymbol{x}) = \boldsymbol{x} + \int_0^T \boldsymbol{v}_\theta(\boldsymbol{x}_t, t)\mathrm{d}t, \quad \boldsymbol{x}_1 = \boldsymbol{x}. \tag{9}$$

which induces an instantaneous change-of-variable formula of parametric log-density: $\log p_t(\boldsymbol{x}_t) = \log p_0(\boldsymbol{x}_0) - \int_0^t \nabla \cdot \boldsymbol{v}_\theta(\boldsymbol{x}_t, s)\mathrm{d}s$. Although the formulation is continuous, any practical evaluation

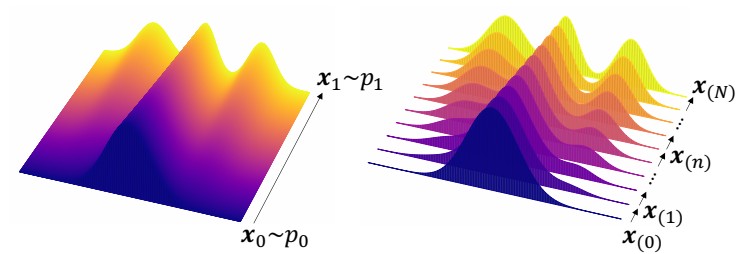

Figure 1: The illustration of continuous-time flow (left) and discrete-time block flow (right).

of (9) inevitably relies on a *discrete* time grid; indeed, a single residual block can be viewed as a forward–Euler step that integrates the same ODE for one unit time. Conversely, this observation motivates us to utilize the benefit of continuous-time NF (neural ODE) inside the discrete-time NF framework by setting the $n$-th block $f_n$ to be a neural ODE on a sub-interval of time.

Specifically, let the time horizon $[0, T]$ be discretized into $N$ subintervals $[t_{n-1}, t_n]$ and $\boldsymbol{x}_t$ solves the ODE with respect to the velocity field $v_\theta(\boldsymbol{x}_t, t)$. The $n$-th block mapping (associated with the subinterval $[t_{n-1}, t_n]$) is defined by the integral flow:

$$\boldsymbol{x}_{(n)} = \boldsymbol{x}_{(n-1)} + \int_{t_{n-1}}^{t_n} \boldsymbol{v}_\theta(\boldsymbol{x}_t, t)\mathrm{d}t, \quad \boldsymbol{x}_{t_{n-1}} = \boldsymbol{x}_{(n-1)}. \tag{10}$$

Equation 10 inherits the analytically tractable change-of-variables formula of CNFs—one integrates $\nabla \boldsymbol{v}_\theta$ over the same interval.

Obviously, standard FM training treats the entire trajectory on $[0, T]$ as a monolithic map and optimizes all $N$ blocks jointly with a single objective (typically maximum likelihood), see Fig. 1(left). While by adopting a flow sub-network inside each residual block, one can design a discrete-time flow model that is free-form, automatically invertible (by using small time step to ensure sufficiently accurate numerical integration of the ODE such that the ODE trajectories are distinct), and enjoys the same computational and expressive advantage as continuous-time flow policy. If using $\boldsymbol{v}_\theta$ on $[t_{n-1}, t_n]$, the $n$-th block can potentially be trained independently and progressively, meaning that only one block is trained at a time and the $n$-th block is trained only after the previous $(n-1)$ blocks are fully trained and fixed. We call such flow implementing the iterative steps the iterative flow, and the key of our SWFP is to design a step-wise loss to train each block.

To solve the above problem, we first model the continuous-time diffusion process by a partial differential equations (PDE), i.e., the Fokker-Planck equation,

$$\frac{\partial \rho(t, \boldsymbol{x})}{\partial t} = \nabla \cdot (\nabla U(\boldsymbol{x})\rho(t, \boldsymbol{x})) + \beta \nabla^2 \rho(t, \boldsymbol{x}), \tag{11}$$

describes the time evolution of the distribution $\rho$ of a set of particles undergoing drift and diffusion. Note that the density evolution through the CE (Eq. 2) and the FPE (Eq. 11) are mathematically equivalent when we set $\boldsymbol{v}_t(\boldsymbol{x}) = -\nabla U(\boldsymbol{x}) - \nabla \log \rho(t, \boldsymbol{x})$.

Motivated by the work of Jordan, Kinderlehrer, and Otto (Jordan et al., 1998; Mokrov et al., 2021), related diffusion processes to energy-minimizing trajectories in the Wasserstein space, providing a discrete-time counterpart of the transport process described by Eq. 11, the JKO scheme.

The classical JKO scheme computes a sequence of distributions $\rho_n, n = 0, 1, \dots$ by

$$\rho_{n+1} = \arg\min_{\rho \in \mathcal{P}} \mathcal{E}(\rho) + \frac{1}{2\tau} \mathcal{W}_2^2(\rho_n, \rho), \tag{12}$$

starting from $\rho_0 \in \mathcal{P}_2$, where $\tau > 0$ controls the step size. $\mathcal{E}$ is an energy functional. The Fokker-Plank equation (11) results from the continuous-time limit (i.e., $\tau \to 0$) of the JKO scheme for the free energy and describes external potentials and local self-interactions:

$$\mathcal{E}(\rho) := \int \rho \log \rho \, dx + \int U(x) \, d\rho(x) \tag{13}$$

which is also expressed as the Kullback-Leibler (KL) divergence to the target distribution $q \propto e^{-U(x)}$.

Strictly speaking, the minimization in eq. (12) is over the Wasserstein-2 space of the density $\rho$, which, in the $n$-th JKO flow block will apply to the pushforwarded density by the mapping in the $n$-th block. Specifically, let $F_{n,\theta}$ denote the forward mapping in the $n$-the block, i.e., $\boldsymbol{x}_n = F_{n,\theta}(\boldsymbol{x}_{n-1}, n-1)$ parameterized by $\boldsymbol{v}_\theta$ over $t \in [t_{n-1}, t_n]$. Denoting the marginal distribution of $\boldsymbol{x}(n)$ by $p_n$, with $p_N = p_1$ (where $\boldsymbol{x}_1$ follows the data distribution), we obtain the density transformation:

$$p_n = (F_{n,\theta})_\# p_{n-1}. \tag{14}$$

**Policy Improvement via JKO Iterations** For maximum entropy reinforcement learning, we instantiate the JKO scheme by setting the target distribution (12) to the energy-based policy from (8). Substituting $U(\boldsymbol{a}) \equiv -Q(\boldsymbol{s}, \boldsymbol{a})/\alpha$ into the energy functional yields the policy optimization objective of the iterative flow policy:

$$\pi_{n+1} = \arg \min_\pi \frac{1}{2\tau} \mathcal{W}_2^2(\pi_n, \pi) + \int \pi \log \pi \mathrm{d}\boldsymbol{a} + \int (-Q(\boldsymbol{s}, \boldsymbol{a})/\alpha)\pi \mathrm{d}\boldsymbol{a} \tag{15}$$

This objective is minimized over the parameters $\theta$ of the flow map $F_\theta$, which transports the distribution $\pi_n$ to $\pi_{n+1}$. The loss has a clear interpretation: it seeks a new policy $\pi_{n+1}$ that minimizes the Max-Entropy objective while staying proximal to the previous policy $\pi_n$ in the Wasserstein space. The Wasserstein term serves as the policy distance $d$ in (6) to regularize the "amount of movement" from the current density $p_{\pi_{n-1}}$ by the transport map $F_{n,\theta}$. We provide more analysis in Appendix B.

**Proposition 4.1** *For an iterative flow with the energy functional defined in (15), $\pi(\boldsymbol{a}|\boldsymbol{s})$ converges to $p_{s,\pi}(\boldsymbol{a}) \propto e^{Q(\boldsymbol{a},\boldsymbol{s})}$ in the infinite-time limit with $Q(\boldsymbol{a}, \boldsymbol{s})$ satisfying the following modified Bellman equation:*

$$Q(\boldsymbol{a}, \boldsymbol{s}) = r(\boldsymbol{a}, \boldsymbol{s}) + \gamma \mathbb{E}_{\boldsymbol{s}' \sim \rho_\pi} \left[ V_\pi(\boldsymbol{s}') - \mathcal{H}(\pi(\cdot|\boldsymbol{s}')) \right]$$

*where $V_\pi(\boldsymbol{s}') \triangleq \log \int_{\mathcal{A}} \exp(Q(\boldsymbol{a}, \boldsymbol{s}'))\mathrm{d}\boldsymbol{a}$.*

The proof is given in Appendix 4.1. This result provides theoretical grounding that by iteratively minimizing the JKO objective, SWFP converges to the optimal MaxEnt policy, ensuring stability through the intrinsic Wasserstein trust region imposed by the JKO operator. This step-wise form essentially decomposes the pushforward process of a flow model regarded as a policy into finite blocks of policy probability transport $(F_{n,\theta})_\#(\pi_{n-1})$.

### 4.3 STEPWISE POLICY OPTIMIZATION FOR FLOW MATCHING

From the JKO perspective, a policy $\pi$ is viewed as a point on the Wasserstein manifold of probability measures, where optimization involves following the steepest ascent of reward while paying a transportation cost measured by the W2 metric. Rather than learning one monolithic flow that jumps directly from the behaviour distribution to the optimum, we decompose the trajectory into a sequence of short, local moves. Each flow-matching sub-model is trained to transport the current policy distribution to the next JKO iterate, so the global optimisation is realised as a chain of conservative, well-conditioned steps. This stepwise decomposition maintains proximal distributions between consecutive policy iterates, enabling stable and efficient optimization. Building upon this geometric foundation, we now present a practical RL algorithm (see Algorithm 1 in Appendix).

**Soft critic for target policy** Equation (15) adopts the JKO scheme to optimize the policy $\pi$ by particle approximation, i.e., $\pi \propto \frac{1}{M} \sum_{i=1}^M \delta_{\boldsymbol{a}^i}$. However, it is difficult due to the infinite time horizon and the unknown reward function $r(\boldsymbol{s}, \boldsymbol{a})$ when calculating $Q(\boldsymbol{a}^i, \boldsymbol{s})$. To address this, we approximate the soft Q-function, $Q(\cdot, \boldsymbol{s})$, with a deep neural network $Q_s^\varphi(\boldsymbol{s}, \boldsymbol{a})$ parametrized by $\varphi$, i.e., $p_{\boldsymbol{s},\pi}(\boldsymbol{a}) \propto e^{Q_s^\varphi(\boldsymbol{s},\boldsymbol{a})}$. The neural network $Q_s^\varphi(\boldsymbol{s}, \boldsymbol{a})$ naturally leads to a soft approximation of the standard Q-function according to Proposition 4.1.

We optimize the Q-network using the Bellman error as in the soft-Q learning setting. Specifically, in each iteration, we optimize the following objective function:

$$J_Q(\varphi) \triangleq \mathbb{E}_{\boldsymbol{s}_t \sim q_{\boldsymbol{s}_t}, \boldsymbol{a}_t \sim q_{\boldsymbol{a}_t}} \left[ \frac{1}{2} \left( \hat{Q}_s^{\bar{\varphi}}(\boldsymbol{s}_t, \boldsymbol{a}_t) - Q_s^\varphi(\boldsymbol{s}_t, \boldsymbol{a}_t) \right)^2 \right], \tag{16}$$

where $q_{s_t}$ and $q_{a_t}$ are arbitrary distributions with support on $\mathcal{S}$ and $\mathcal{A}$, respectively;

$$\hat{Q}_s^{\bar{\varphi}}(s_t, a_t) = r(s_t, a_t) + \gamma \mathbb{E}_{s_{t+1} \sim \rho_\pi} \left[ V_s^{\bar{\varphi}}(s_{t+1}) \right] \tag{17}$$

is the target Q-value, with

$$V_s^{\bar{\varphi}}(s_{t+1}) = \log \mathbb{E}_{q_{a'}} \left[ \frac{\exp(Q_s^{\bar{\varphi}}(s_{t+1}, a'))}{q_{a'}(a')} \right] - \mathcal{H}(q_{a'}); \tag{18}$$

$\bar{\varphi}$ represents the parameters of the target Q-network, as used in standard deep Q-learning. $q_{a_t}$ can be set to the distribution induced by the sampling network, built upon the flow model framework. Then, given $Q_s^{\varphi}$, we could adopt the particle approximation with the the JKO scheme to optimize the policy.

**Parallel block training** Under the JKO scheme, we specify a discrete time step schedule $\gamma_n = 1/N$ for simplicity on the time interval $[0, 1]$. Suppose the data distribution $p(a_1)$ has a regular density $q$. Starting from the noise distribution $p_0$ (when $n = 1$), we need recursively construct target density $p_n = F_{n,\theta,\#} p_{n-1}$, because of the dependency of $\pi_n$ on $\pi_{n-1}$ in (15). In the first iteration, we train the first velocity $v_\theta(a, 0)$. After the first block is trained, the particle positions are updated using the learned transport map $F_{1,\theta}$. In the next iteration, we train the velocity field $v_\theta(a, t_1), t_1 = \gamma * 1$, and the initial position of the particles are $a^{(i)}(t_1)$ which have been computed from the previous iteration. This proce-

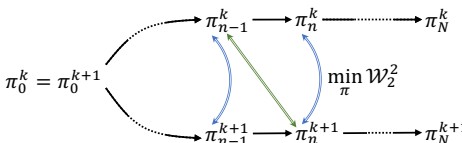

Figure 2: Illustration of SWFP's parallel block training. Each horizontal chain of black arrows represents the forward composition of flow blocks within a training epoch. Double-headed arrows indicate the Wasserstein distance minimization objectives: $\mathcal{W}_2^2(\pi_{n-1}^{k+1}, \pi_{n-1}^k)$ and $\mathcal{W}_2^2(\pi_{n-1}^k, \pi_n^{k+1})$ during optimization of block $n$.

dure continues for $n = 1, 2, \ldots, N$ for $N$ steps. This sequential training approach would clearly be inefficient and would require either independent or specially designed networks to mitigate catastrophic forgetting. Therefore, we propose parallel block training by introducing the triangle inequality of the Wasserstein distance: $\mathcal{W}_2^2(\pi_{n-1}^{k+1}, \pi_n^{k+1}) \leq \mathcal{W}_2^2(\pi_{n-1}^{k+1}, \pi_{n-1}^k) + \mathcal{W}_2^2(\pi_{n-1}^k, \pi_n^{k+1})$. Specifically, during training at the $k + 1$-th epoch, we utilize the old particle positions to compute the iterative process while additionally introducing the regularization term $W_2^2(\pi_n^k, \pi_n^{k+1})$:

$$\pi_{n+1}^{k+1} = \arg \min_{\hat{\pi}} \frac{1}{2\tau} \left( \mathcal{W}_2^2(\pi_n^k, \hat{\pi}) + \mathcal{W}_2^2(\pi_{n+1}^k, \hat{\pi}) \right)$$
$$+ \int \hat{\pi} \log \hat{\pi} \mathrm{d}a + \int (-Q(s, a)/\alpha) \hat{\pi} \mathrm{d}a \triangleq J_{\pi_n} \tag{19}$$

The overall transport from the initial policy $\pi_0$ to the target policy is decomposed into $N$ consecutive FM blocks $\{F_{n,\theta}\}_{n=1}^N$. At block $n$ the particles $\{a_{(n)}^{(i)}\}_{i=1}^M$ is pushed forward by $a_{(n+1)}^{(i)} = F_{n,\theta}(a_{(n)}^{(i)}, s, n)$ (see Appendix E).

Using the chain rule,

$$\frac{\partial J_{\pi_n}^\theta}{\partial \theta} = \mathbb{E} \left[ \frac{\partial J_{\pi_n}^\theta}{\partial a_{(n)}^i} \frac{\partial a_{(n)}^i}{\partial \theta} \right],$$

so $\theta$ can be updated with standard stochastic gradient descent; $\partial a_{(n)}^i / \partial \theta$ is obtained by back-propagating through $F_{n,\theta}$, and $\partial J_\pi^\theta / \partial a_{(n)}^i$ follows directly from the loss. Repeating this block-wise procedure yields controlled step sizes in Wasserstein space and ensures monotonic improvement of the entropy-regularised return.

## 5 EXPERIMENTAL RESULTS

To comprehensively assess the proposed SWFP, we organize the experiments along four complementary axes that move from toy and RL benchmark comparison to a systematic hyper-parameter

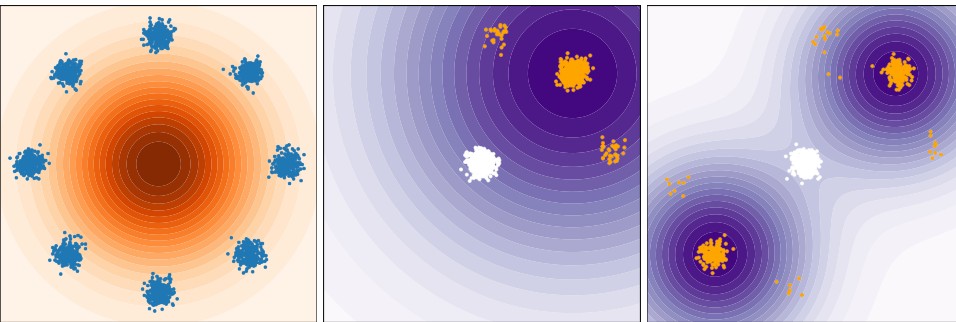

Figure 3: **Illustration of SWFP performance on a bandit toy example:** (left) source distribution and behavior data for pre-training in the replay buffer; (middle, right) implicit SWFP steers the source distribution toward the high-performance behavior policy, heatmap shows target policy.

study. We begin with low-dimensional toy data to make convergence behavior tangible; next, we investigate how quickly the model can adapt in an online, interactive setting; third, we test whether a network pre-trained on large offline datasets can be retargeted to new objectives with minimal effort; and, finally, we analyze the sensitivity of performance to the number of JKO blocks.

## 5.1 FOUNDATIONAL TOY EXPERIMENTS

To evaluate the ability of our discretized step flow model to recover a target distribution, we begin with a synthetic two-dimensional example. The experiment starts with a source distribution as a standard Gaussian centered at the origin, and the behavior data distribution as an eight-component Gaussian mixture (from which 1,000 samples are drawn). The middle and right of Fig. 3 visualize the distribution evolution produced by a six-step SWFP ($N = 6$) as it transports the source distribution towards the target (indicated by the purple level sets). White particles initially sampled from the Gaussian source are gradually transported out of the low-reward annulus and into the high-reward modes, closely following the gradient field of the target policy.

## 5.2 EFFICIENT ONLINE ADAPTATION

We first examine the online adaptation setting, where a diffusion or flow policy pretrained on demonstrations is adapted using online samples. Overall, SWFP consistently and significantly improves the fine-tuning efficiency and asymptotic performance of flow policies. Notably, SWFP maintains high normalized scores across diverse tasks like kitchen-complete-v0 and CALVIN, a stability that contrasts sharply with the considerable fluctuations observed in baselines like DQL and IDQL. It is worth noting that SWFP uses only 5 inference steps, which is significantly fewer than the 100 steps required by other Diffusion methods and also reduces the 20-step setting of the pretrained flow policy. Although this may lead to a performance decline in the early stages of training, it still achieves competitive results by the end of training (e.g., Can). Furthermore, we compared SWFP against Flow Q-Learning (FQL) (Park et al., 2025b), an offline RL method which can also be used for offline-to-online fine-tuning. The FQL policy struggles to match the reward achieved by pure behavior cloning pre-training because gradients flowing to the multi-step policy during one-step distillation interfere with the BC loss. In contrast, SWFP can directly and stably finetunes the multi-step flow matching policy, offering richer representation capabilities than one-step distilled models in FQL. This superior robustness is directly traceable to SWFP's core design. By utilizing the Stepwise JKO Iteration, SWFP ensures that each policy update is intrinsically regularized by the Wasserstein metric. This Wasserstein Trust Region strictly limits the "magnitude of change" in the policy distribution, effectively mitigating catastrophic divergence and compounding errors often seen in off-policy methods operating in high-dimensional continuous action spaces. Furthermore, SWFP breaks the optimization into a chain of short, easily trained flow blocks. This avoids the high computational and stability costs of backpropagating through a monolithic, multi-step flow. In essence, SWFP unifies the expressive power of flow policies with the principled stability of JKO proximal optimization, resulting in a robust and highly effective online fine-tuning approach.

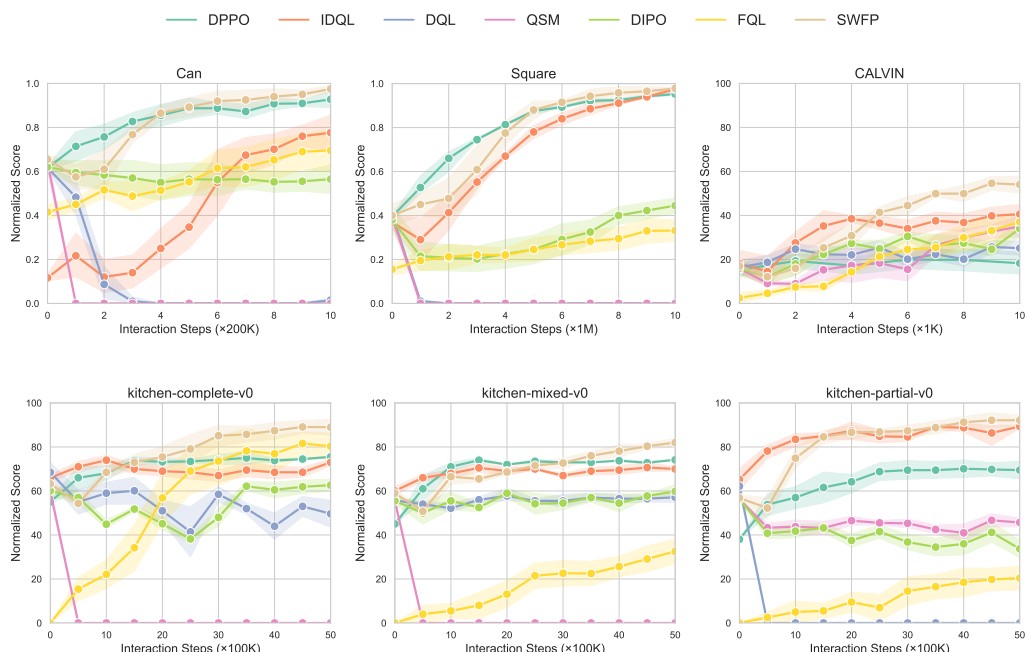

Figure 4: **Learning curves of online fine-tuning with various methods.** Observe that SWFP largely always dominates or attains similar performance to the next best method.

Table 1: Comparison of SWFP with other demo-augmented RL algorithms. SWFP outperforms every other approach, both in terms of the offline performance (left of $\rightarrow$) and performance after fine-tuning (right of $\rightarrow$).

|  | Franka-Kitchen | | | RoboMimic | |
|---|---|---|---|---|---|
|  | Complete-v0 | Mixed-v0 | Partial-v0 | Can-State | Square-State |
| RLPD | $0 \rightarrow 18$ | $0 \rightarrow 14$ | $0 \rightarrow 34$ | $0 \rightarrow 0$ | $0 \rightarrow 3$ |
| Cal-QL | $19 \rightarrow 57$ | $37 \rightarrow 72$ | $59 \rightarrow 84$ | $0 \rightarrow 0$ | $0 \rightarrow 0$ |
| IBRL | $0 \rightarrow 25$ | $0 \rightarrow 13$ | $0 \rightarrow 15$ | $0 \rightarrow 64$ | $0 \rightarrow 50$ |
| SWFP | $63 \rightarrow 89$ | $59 \rightarrow 82$ | $57 \rightarrow 92$ | $65 \rightarrow 97$ | $40 \rightarrow 98$ |

## 5.3 COMPARISON TO DEMO-AUGMENTED RL ALGORITHMS

We investigate the effectiveness of SWFP in the challenging offline-to-online setting, comparing its performance against established methods like RLPD Ball et al. (2023), Cal-QL Nakamoto et al. (2024), and IBRL Hu et al. (2023), all of which leverage offline data for off-policy updates. We evaluate these methods on Franka-Kitchen and RoboMimic environents. All of results are shown on Table 1. Across the Franka-Kitchen domains, SWFP consistently demonstrates superior adaptation capability, achieving impressive score improvements, notably rising from 63 to 89 in Kitchen-Complete-v0 and leading decisively with 92 in Kitchen-Partial-v0—scores where baselines like RLPD and IQL perform significantly worse. This success highlights SWFP's effective mechanism for refining a pretrained flow policy via online interaction. Furthermore, in the RoboMimic environment, SWFP continues to excel, achieving high scores of 97 in Can-State and 98 in Square-State, showcasing its robustness across diverse, high-fidelity scenarios. Al-

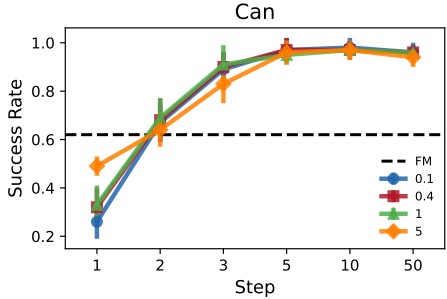

Figure 5: Sensitivity of Hyperparameters.

though IBRL performs the best among the non-flow competitors, a significant performance gap remains, solidifying SWFP as the state-of-the-art approach for fine-tuning expressive flow policies in the offline-to-online paradigm.

## 5.4 SENSITIVITY ANALYSIS

This section investigates the sensitivity of our algorithm to the discrete step number $N$ and the Wasserstein-2 scale. Evaluation on the RoboMimic-Can task over 10 runs yields the results in Figure 5. The figure reveals two main insights: First, increasing the step number leads to a consistent improvement in success rate until performance plateaus at a high level, with a significant jump observed after $N = 5$. Second, and more importantly, the four curves representing different Wasserstein-2 scales $(0.1, 0.4, 1, 5)$ cluster closely together. This indicates that the W-2 trust-region is not a sensitive parameter, as the algorithm maintains strong performance across a wide range of values, consistently outperforming the pre-trained FM baseline (dashed line).

## 6 CONCLUSION

In this work, we introduce the Stepwise Flow Policy (SWFP) framework to enable stable and efficient online fine-tuning of flow-based policies. By aligning the discretized flow-matching process with the JKO scheme from optimal transport, SWFP decomposes policy improvement into a sequence of regularized, local updates in probability space. This approach ensures stable adaptation through Wasserstein trust regions and entropic regularization. Extensive experiments demonstrate that SWFP achieves superior stability and performance compared to existing methods, providing a principled and practical path for online reinforcement learning with generative policies.

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

## A  PROOF OF PROPOSITION 4.1

**Proof A.1** *The first claim of Proposition 4.1 states that the iterated policy $\pi(\mathbf{a}|\mathbf{s})$ converges to $p_{s,\pi}(\boldsymbol{a}) \propto e^{Q(\boldsymbol{a},\boldsymbol{s})}$. Recall that the iterative flow defined in 15 with energy functional in 13 is equivalent to a Fokker-Planck equation:*

$$\partial_n \pi_n = \nabla \cdot (-\pi_n \nabla Q(\boldsymbol{a}, \boldsymbol{s})) + \nabla^2 \pi_n,$$

*where $n$ denotes the iteration index. The Fokker–Planck operator on the right-hand side admits a unique invariant measure. Setting $\partial_n \pi_n = 0$ yields*

$$\nabla \cdot \left( -\pi_\infty \nabla Q \right) + \nabla^2 \pi_\infty = 0.$$

*A standard calculation shows that the unique normalisable solution is the Gibbs measure*

$$\pi_\infty(\boldsymbol{a} \,|\, \boldsymbol{s}) = \frac{1}{Z(\boldsymbol{s})} \exp\big(Q(\boldsymbol{a}, \boldsymbol{s})\big), \qquad Z(\boldsymbol{s}) = \int \exp\big(Q(\boldsymbol{a}, \boldsymbol{s})\big) \, \mathrm{d}\boldsymbol{a}.$$

*Hence, as $n \to \infty$, the sequence $\{\pi_n\}$ converges to $p_{s,\pi}$.*

*For any policy $\pi$ the soft state–action value is defined as*

$$Q_\pi(\boldsymbol{s}_t, \boldsymbol{a}_t) \;=\; r(\boldsymbol{s}_t, \boldsymbol{a}_t) + \; \mathbb{E}_{(\boldsymbol{s}_{t+1:t+\infty}, \boldsymbol{a}_{t+1:t+\infty}) \sim (\rho_\pi, \pi)} \Big[ \sum_{l=1}^{\infty} \gamma^l r(\boldsymbol{s}_{t+l}, \boldsymbol{a}_{t+l}) \Big].$$

*Isolating the first transition yields*

$$Q_\pi(\boldsymbol{s}_t, \boldsymbol{a}_t) = r(\boldsymbol{s}_t, \boldsymbol{a}_t)$$

$$+ \; \gamma \, \mathbb{E}_{\boldsymbol{s}_{t+1} \sim \rho_\pi} \, \mathbb{E}_{\boldsymbol{a}_{t+1} \sim \pi} \Big[ r(\boldsymbol{s}_{t+1}, \boldsymbol{a}_{t+1}) + \mathbb{E}_{(\boldsymbol{s}_{t+2:\infty}, \boldsymbol{a}_{t+2:\infty})} \sum_{l=1}^{\infty} \gamma^l r(\boldsymbol{s}_{t+1+l}, \boldsymbol{a}_{t+1+l}) \Big].$$

*Introduce the (soft) state value $V_\pi(\boldsymbol{s}) = \mathbb{E}_{\boldsymbol{a} \sim \pi}\big[ Q_\pi(\boldsymbol{s}, \boldsymbol{a}) \big]$, and recall that the Boltzmann policy can be written as $\pi(\boldsymbol{a} \mid \boldsymbol{s}) = \exp\big( Q_\pi(\boldsymbol{s}, \boldsymbol{a}) - V_\pi(\boldsymbol{s}) \big)$. Using this form and the definition of (differential) Shannon entropy $\mathcal{H}(\pi(\cdot \mid \boldsymbol{s})) = -\mathbb{E}_{\boldsymbol{a} \sim \pi}\big[ \log \pi(\boldsymbol{a} \mid \boldsymbol{s}) \big]$, we can rewrite the inner expectation:*

$$\mathbb{E}_{\boldsymbol{a}_{t+1} \sim \pi}\big[ Q_\pi(\boldsymbol{s}_{t+1}, \boldsymbol{a}_{t+1}) \big] = V_\pi(\boldsymbol{s}_{t+1}) - \mathcal{H}(\pi(\cdot \mid \boldsymbol{s}_{t+1})).$$

*Substituting this result into the previous expansion gives the soft Bellman equation*

$$Q_\pi(\boldsymbol{s}_t, \boldsymbol{a}_t) = r(\boldsymbol{s}_t, \boldsymbol{a}_t) + \gamma \, \mathbb{E}_{\boldsymbol{s}_{t+1} \sim \rho_\pi} \Big[ V_\pi(\boldsymbol{s}_{t+1}) - \mathcal{H}(\pi(\cdot \mid \boldsymbol{s}_{t+1})) \Big].$$

■

## B   More Analysis

The per-step training objective, defined by Equation 15, can be viewed as the addition of two key components: the variational objective (measuring the closeness of the pushforwarded density to the target policy) and the Wasserstein regularization term. This $\mathcal{W}_2$ term plays a critical role by serving as a proximal operator that actively regularizes the "amount of movement" enforced by the transport map $F_{n,\theta}$. Intuitively, among all transport maps that can successfully reduce the KL divergence to the target policy, the regularization selects the one that achieves the result with the smallest possible movement. By penalizing excessive displacement between consecutive policy iterates, the $\mathcal{W}_2$ term encourages straighter, more stable transport paths in the probability space (Xie & Cheng, 2025). This stabilization inherently reduces the risk of mode collapse and may decrease the total number of neural network blocks required to reach the final target distribution. Critically, in a particle-based implementation of the $n$-th flow block, this $\mathcal{W}_2$ term becomes computationally tractable, directly corresponding to the average squared movement over $m$ particles along the ODE trajectory: $\frac{1}{m} \sum_{i=1}^{m} \|x^i(t_n) - x^i(t_{n-1})\|^2$.

The convergence guarantee for the SWFP is established by mirroring the analysis of W2-proximal gradient descent (GD), leveraging the assumption that each JKO minimization step is solved with an approximation error of $O(\varepsilon)$. By utilizing the $\lambda$-convexity of the KL-divergence functional, $\mathcal{F}(\rho) := \text{KL}(\rho \| q)$, in the Wasserstein space, we derive the Evolution Variational Inequality (Muratori & Savaré, 2020; Xie & Cheng, 2025):

$$\left( 1 + \frac{\tau \lambda}{2} \right) \mathcal{W}^2(p_{n+1}, q) + 2\tau \left( \mathcal{F}(p_{n+1}) - \mathcal{F}(q) \right) \leq \mathcal{W}^2(p_n, q) + O(\varepsilon^2),$$

which is a key step that confirms the exponential convergence of the Wasserstein GD, ensuring the policy iterates $p_n$ quickly approach the target distribution $q$. Specifically, this analysis shows that a KL-divergence error of $O(\varepsilon^2)$ is achieved after executing only approximately $\log(1/\varepsilon)$ JKO steps, demonstrating the theoretical efficiency of SWFP in recovering the target policy.

## C   Details for Experiments

All experiments are conducted on an NVIDIA Tesla A100 80GB GPU, and all fine-tuning methods use the same pre-trained policy. In the pretraining, the observations and actions are normalized to

$[0, 1]$ using min/max statistics from the pre-training dataset. No history observation (pixel, proprioception, or ground-truth object states) is used. The deffusion policy and flow policy is trained with learning rate $1e-4$ decayed to $1e-5$ with a cosine schedule, weight decay 1e-6 and 50 parallelized. For Franka-Kitchen and Robomimic tasks, epochs is 8000 and batch size is 128; for CALVIN tasks, epochs is 5000 and batch size is 512.

## C.1 Details and Hyper-parameters for SWFP

SWFP has two most important hyperparameters: the discretized stepsize $\tau$ and the temperature $\alpha$. Also note that we can only evaluate the gradient of W2 up to a constant, there needs to a parameter balancing the gradient of the energy functional $F$ and the Wasserstein term. We denote this hyparameter as $\epsilon$. In the experiments, if not explicitly stated, the default setting for these parameters are $\epsilon = 0.4$. For temperature parameter, we performed a grid search over $\alpha \in \{2, 3, 4, 5, 6, 7, 8\}$.

Table 2: Hyper-parameters for SWFP

| Parameter | Task | | | |
| --- | --- | --- | --- | --- |
| | Franka-Ketichen | CALVIN | Robomimic-Can | Robomimic-Square |
| Buffer Size | 1000000 | 250000 | 250000 | 250000 |
| Actor Learning Rate | 1.00E-05 | 1.00E-05 | 1.00E-05 | 1.00E-05 |
| Discount Factor | 0.99 | 0.99 | 0.999 | 0.999 |
| Optimizer | | Adam | | |
| ODESolver | | Midpoint Euler | | |
| Block Number $N$ | 5 | 5 | 5 | 5 |
| $\alpha$ | 5 | 5 | 4 | 4 |
| $\tau$ | 0.1 | 0.1 | 0.1 | 0.1 |
| Actor Batch Size | 1024 | 1024 | 1024 | 1024 |
| Hidden Layer Sizes | [512, 512, 512] | [512, 512, 512] | [256, 256, 256] | [256, 256, 256] |
| Q Batch Size | 256 | 256 | 256 | 256 |

## C.2 Details and Hyper-parameters for Baselines

**DPPO** For the state-based tasks Robomimic and FrankaKitchen, we trained DPPO-MLP following the original paper's specifications, using an action chunking size of 4 for Robomimic and 8 for FrankaKitchen. For the pixel-based task CALVIN, we trained DPPO-ViT-MLP with an action chunking size of 4.

**IDQL** We employ the IDQL-Imp version of IDQL, wherein the Q-function, value function, and diffusion policy are refined through new experiences. For Robomimic tasks, we employ the same network architecture as DPPO, while the original IDQL architectures are preserved for Franka-Kitchen and CALVIN. For the IQL $\tau$ expectile, we set it to 0.7 for each task. For network architectures, we use the default residual multilayer perception (MLP) (three blocks of [256, 1024, 256]-sized residual layers) for the behavioral diffusion policy.

**DQL** We set the weighting coefficient to 0.5 for Robomimic, 0.005 for Franka-Kitchen and 0.01 for CALVIN.

**IBRL** We adhere to the original implementations' hyperparameters, with wider (1024) MLP layers and dropout during pre-training.

**Cal-QL** We set the mixing ratio to 0.25 for Franka-Kitchen and 0.5 for CALVIN and Robomimic.For the network size, we consider both [256, 256, 256]- and [512, 512, 512]-sized MLPs.

**QSM** We set the weighting coefficient $\alpha$ to 10.

**DIPO** We set the weighting coefficient $\alpha$ to $1e-4$ and hyperparameter $M$ to 10.

**FQL** For reproducibility and a fair comparison against FQL, we adopted the following setup based on Zhang et al. (2025) and the official literature (Park et al., 2025b). We set the number of offline

---

**Algorithm 1** Online Policy Training (SWFP)

---

**Require:** The pre-trained flow policy $\pi^\theta$, replay buffer $\mathcal{B} = \varnothing$, initial Q-function $Q^\varphi$, MDP $\mathcal{M}$.
1: Target parameters: $\bar{\theta} \leftarrow \theta$, $\bar{\varphi} \leftarrow \varphi$
2: **for** each epoch $k$ in $\{1, 2, \ldots, K\}$ **do**
3:    % Sampling and experience replay.
4:    Interact with $\mathcal{M}$ using the policy $\pi^{\bar{\theta}}$.
5:    Update replay buffer $\mathcal{B}$.
6:    Sample minibatch $(\boldsymbol{s}, \{\boldsymbol{a}_n\}_{n=0}^N, r, \boldsymbol{s}')$ from $\mathcal{B}$.
7:    % Update Q function
8:    Compute empirical values $\hat{V}^{\bar{\varphi}}(\boldsymbol{s}')$.
9:    Update $\varphi$ with the empirical gradient $\hat{\nabla}_\varphi J_Q(\varphi)$.
10:   % Update policy
11:   **for** each policy improvement step **do**
12:     Sample $n$ uniformly from $\{1, 2, \ldots, N\}$.
13:     Compute $\mathcal{W}_2^2(\pi_n^\theta, \pi_{n-1}^{\bar{\theta}})$ and $\mathcal{W}_2^2(\pi_n^\theta, \pi_n^{\bar{\theta}})$.
14:     Compute the empirical gradient $\hat{\nabla}_\theta J_{\pi_n}(\theta)$
15:     Update prior policy parameters $\theta$.
16:   **end for**
17:   Update target Q-function parameters $\bar{\varphi}$.
18: **end for**

---

pre-training steps such that the total sample consumption during the offline phase is no less than the pre-trained consumption of DPPO or SWFP. The code and hyperparameter settings for FQL were referenced from the official papers. The temperature coefficient $\alpha_{\text{FQL}}$ is the most critical hyperparameter in FQL. We followed the instructions of the original paper and performed a hyperparameter scan for $\alpha_{\text{FQL}}$ in the range $[0.03, 0.1, 0.3, 1, 3, 10]$ to obtain the optimal value for the policy update. The Behavior Cloning coefficient was set to 3.0, and the actor/critic scheduler warmup was set to 10.

# D  STATEMENT ON LARGE LANGUAGE MODEL USAGE

This paper employed Large Language Models to assist in the writing process. The LLM was used exclusively for the purpose of language polishing, which included:

- Correcting grammatical errors.
- Improving sentence fluency and readability.
- Refining word choice for better academic tone.

The LLM was **not** used for generating original ideas, formulating research hypotheses, conducting data analysis, or interpreting results. All intellectual content and scholarly contributions are solely those of the authors. The authors have thoroughly reviewed, revised, and take complete responsibility for the entire content of this manuscript.

# E  PUSHFORWARDING PARTICLES.

Specially, the minimization of (the empirical version of 15 and 19 in each step of the JKO flow is ready to be computed on particles $\{\boldsymbol{a}^{(i)}\}_{i=1}^M$, namely the finite data samples. At a time step $t_n$, the particle positions correspond to training samples $\boldsymbol{a}_{(n)}^{(i)}$. We train the $n$-th block, and the velocity field $v_\theta(\boldsymbol{a}, n)$ over the interval $t \in [t_n, t_{n+1}]$ is modeled by a neural network with parameters $\theta$. The empirical version of 15 gives the training objective of the $n$-th block as

$$\min_\theta \frac{1}{M} \sum_{i=1}^M \left( -\frac{Q(\boldsymbol{s}, \boldsymbol{a}^{(i)}(t_{n+1}))}{\alpha} - \int_{t_n}^{t_{n+1}} \nabla \cdot \boldsymbol{v}_\theta(\boldsymbol{a}^{(i)}(t; \theta), t) dt \right) + \frac{1}{2\tau M} \sum_{i=1}^M \|\boldsymbol{a}^{(i)}(t_n) - \boldsymbol{a}^{(i)}(t_{n+1})\|^2,$$

$$(20)$$

where $\gamma > 0$ controls the step size. After the $n$-th block is trained, the particle positions are updated using the learned transport map 10 on $[t_n, t_{n+1}]$, namely,

$$\begin{cases} \boldsymbol{a}^{(i)}(t_{n+1}) = \boldsymbol{a}^{(i)}(t_n) + \int_{t_n}^{t_{n+1}} \boldsymbol{v}_\theta(\boldsymbol{a}^{(i)}(t), t)dt, \\ \dot{\boldsymbol{a}}^{(i)}(t_n) = \boldsymbol{v}_\theta(\boldsymbol{a}^{(i)}(t_n), t_n). \end{cases} \tag{21}$$

For the empirical version of 19, we utilize the old particle positions to compute the regularization term: $\|\boldsymbol{a}^{k-1,(i)}(t_n) - \boldsymbol{a}^{k,(i)}(t_{n+1})\|^2 + \|\boldsymbol{a}^{k-1,(i)}(t_{n+1}) - \boldsymbol{a}^{k,(i)}(t_{n+1})\|^2$.

# F    ADDITIONAL EXPERIMENTS

To demonstrate the robustness and scalability of the Stepwise Flow Policy (SWFP) across diverse domains, and to address the reviewer's request for locomotion benchmarks and a direct comparison with Flow Q-Learning (FQL), we conducted extensive experiments on five challenging tasks from the **OGBench suite** Park et al. (2025a). The results below validate the effectiveness of SWFP in both the pure Offline RL and the Offline-to-Online Fine-tuning settings.

## F.1    OFFLINE REINFORCEMENT LEARNING RESULTS

Table 3 presents the performance of SWFP and various established implicit and flow-based offline RL baselines on the OGBench suite. The code is from Flow Q-learning (Park et al., 2025b). The results confirm that the SWFP framework, even when restricted to the static dataset, learns a highly effective policy. We use the following 7 methods as representative examples of a variety of algorithm types and policy extraction strategies. Flow advantage-weighted actor-critic (FAWAC) is a flow variant of AWAC (Nair et al., 2020), which uses AWRas the policy learning objective. Flow behavior-regularized actor-critic (FBRAC) is the flow counterpart of Diffusion-QL (DQL)based on the naive Q loss with backpropagation through time. Implicit flow Q-learning (IFQL) is the flow counterpart of IDQL based on rejection sampling. SWFP achieves performance comparable to the best-in-class flow method, FQL, by matching the peak score on `antmaze-large` (**80**) and reaching a competitive score on `humanoidmaze-medium`. It also achieves the highest score on `antsoccer-arena` (**45** vs. FQL's 40). This strong offline performance corroborates that the JKO-based policy representation is highly effective at capturing high-quality actions while implicitly regularizing against distributional shift.

## F.2    ONLINE FINE-TUNING RESULTS

Table 4 highlights the critical online performance, which tests the stability and efficacy of the RL fine-tuning phase. Scores are presented as $Initial \rightarrow Final$ success rates.

The results strongly support the core thesis of our paper—that SWFP provides a superior framework for online refinement of flow policies, especially compared to FQL:

- **Superior Final Performance:** SWFP achieves the **highest final success rate** or is tied for first in **5** out of **5** tasks, demonstrating its overall robustness and strong policy convergence. When directly compared to FQL, SWFP consistently achieves a slightly higher or equal final score.

- **Stability:** The successful and large improvement in success rates (e.g., $10 \rightarrow 84$ in `antsoccer-arena`) demonstrates that the SWFP update, grounded in the JKO scheme, provides the stable policy gradients necessary to move the policy far beyond the initial BC performance.

Table 3: Offline RL results on OGBench Park et al. (2025a).

| Task | BC | IQL | IDQL | FAWAC | FBRAC | IFQL | FQL | SWFP |
|------|-----|-----|------|-------|-------|------|-----|------|
| antmaze-large | 12 | 53 | 21 | 6 | 60 | 30 | **80** | **80** |
| antmaze-giant | 0 | 4 | 0 | 0 | 4 | 3 | 4 | **9** |
| humanoidmaze-medium | 1 | 33 | 1 | 19 | 38 | **60** | 58 | 55 |
| humanoidmaze-large | 0 | 2 | 1 | 0 | 2 | **10** | 5 | 4 |
| antsoccer-arena | 1 | 8 | 12 | 12 | 16 | 29 | 40 | **45** |

Table 4: Online Finetuning RL results on OGBench Park et al. (2025a).

| Task | IDQL | FBRAC | IFQL | FQL | SWFP |
|------|------|-------|------|-----|------|
| antmaze-large | $12 \rightarrow 18$ | $13 \rightarrow 33$ | $10 \rightarrow 30$ | $13 \rightarrow 84$ | $13 \rightarrow \mathbf{86}$ |
| antmaze-giant | $0 \rightarrow 0$ | $0 \rightarrow 0$ | $0 \rightarrow 0$ | $0 \rightarrow \mathbf{16}$ | $0 \rightarrow \mathbf{16}$ |
| humanoidmaze-medium | $1 \rightarrow 0$ | $8 \rightarrow 8$ | $8 \rightarrow 60$ | $8 \rightarrow 50$ | $8 \rightarrow \mathbf{60}$ |
| humanoidmaze-large | $0 \rightarrow 0$ | $0 \rightarrow 0$ | $0 \rightarrow 0$ | $0 \rightarrow 10$ | $0 \rightarrow \mathbf{15}$ |
| antsoccer-arena | $1 \rightarrow 6$ | $10 \rightarrow 8$ | $12 \rightarrow 33$ | $10 \rightarrow 80$ | $10 \rightarrow \mathbf{84}$ |

