# OpenReview forum: "Iterative Refinement of Flow Policies in Probability Space for Online Reinforcement Learning"
_ICLR.cc/2026/Conference — Submitted to ICLR 2026_

### Official Review · Reviewer_sK8u · 2025-10-23

**Soundness:** 2
**Presentation:** 1
**Contribution:** 2
**Rating:** 2
**Confidence:** 3

**Summary:**

This paper presents a framework for fine-tuning a flow-based policy.
In the proposed framework, the transport from the actions generated by an initial policy to those of the optimal policy is modeled based on the Jordan–Kinderlehrer–Otto (JKO) scheme.
The optimal policy is approximated by a Boltzmann distribution derived from entropy-regularized reinforcement learning (RL).
The transport from the initial policy to the optimal policy is modeled in a step-wise manner; that is, when the flow from the initial policy to the optimal policy is discretized into $N$ steps, $N$ models are trained to estimate each transition.

The proposed method was evaluated on manipulation tasks in the Franka-Kitchen and RoboMimic environments, where it outperformed the baseline methods.

**Strengths:**

- Modeling the transport from the initial policy to the optimal policy for policy refinement appears to be novel.

- The experiments demonstrate the advantages of the proposed approach.

**Weaknesses:**

- The presentation lacks clarity, and the paper is not well-organized.

- The experimental evaluation is limited and requires additional experiments.

If I understand correctly, the transport from the actions generated by the initial policy to those of the optimal policy is approximated using step-wise models. However, I am not fully certain whether this interpretation is correct—please clarify if my understanding is mistaken.

While the JKO scheme is introduced in the paper, the update rule appears to correspond to a standard entropy-regularized RL objective function with a Wasserstein distance–based regularization term. It is therefore unclear why the JKO scheme is emphasized. Are the actions generated by the initial policy pushed forward through a process similar to Stein variational gradient descent?
In addition, the procedure for solving the objective in Equation (19) is not clearly described, which makes it difficult to reproduce the proposed method.

Although the proposed approach was evaluated on manipulation tasks (Franka-Kitchen and RoboMimic), evaluations on locomotion benchmarks such as D4RL or OGBench are missing. Since algorithmic behavior often differs between manipulation and locomotion domains, I recommend including additional experiments on locomotion tasks.

Moreover, Flow Q-Learning is now a well-known and relevant baseline, as it also employs a flow-based policy and addresses offline-to-online RL problems. A comparison with Flow Q-Learning [R1] would therefore strengthen the paper.

[R1] Flow Q-Learning. Seohong Park, Qiyang Li, Sergey Levine. ICML 2025.

*Minor Comments*

- Line 207: “can written as” → “can be written as”

- Line 114: “dynamic of” → “dynamics of”

**Questions:**

- Please confirm whether my understanding is correct: is the transport from the actions generated by the initial policy to those of the optimal policy approximated by step-wise models?

- Please explain in detail how the objective in Equation (19) is solved in the proposed method.

- Please include comparisons with Flow Q-Learning and experiments on locomotion tasks.

**Details Of Ethics Concerns:**

I am also reviewing Submission 691, titled “Value Gradient Flow: Behavior Regularized RL without Regularization.”
I noticed that the idea presented in this paper is highly similar to that of Submission 10969. Both submissions share the same main approach: a flow policy is initialized using a given dataset to model the behavior policy, and the transport from the actions generated by the initial policy to those of the optimal policy is learned. The optimal transport is modeled using the Jordan Kinderlehrer Otto (JKO) scheme.

There are, however, some differences between the two submissions. In Submission 691, the update rule, which is analogous to Stein Variational Gradient Descent, is clearly described, and the regularization based on limiting the transport budget is explained. In addition, the gradient flow appears to be modeled using a single model. The method is evaluated on both locomotion and large language model (LLM) fine tuning tasks.
In contrast, in Submission 10969, the update rule is not clearly described, and the transport is modeled with step wise models. The method is evaluated on manipulation tasks.

In my view, the differences between these two papers are minimal. If they are authored by the same group, these submissions may constitute a dual submission of essentially the same work. However, if the authors are different, then this overlap is less concerning.
As I do not have access to the author identities, I am simply raising a flag to alert the Area Chairs and Program Chairs to this potential issue. I will also raise a similar flag for Submission 691.

---

> ### Author Response · Authors · 2025-11-21
>
> We greatly appreciate the reviewer's thorough and insightful feedback. We have revised the Introduction, Related Work, and Results sections to improve clarity and organization.
>
> ### **W1/Q1: Transport Interpretation (Correctness Confirmation)**
>
> Yes, your interpretation is fundamentally correct.
> The Stepwise Flow Policy (SWFP) views the entire flow generation process (from $x_0 \sim \mathcal{N}(0, I)$ to the final action $\mathbf{a}$) as an optimal transport path from a simple prior distribution to the complex action distribution.The core idea is that the standard **Euler discretization** used to solve the flow ODE, when viewed in the probability space, corresponds exactly to a series of steps in the JKO scheme. The JKO scheme models a continuous flow of probability distributions as a sequence of discrete, localized optimal transport steps. Therefore, the policy improvement is achieved by iteratively refining the velocity field $v_{\theta}$, pushing the action distribution toward the reward-weighted optimal one.
>
> ### **W2/Q2: Emphasis on JKO and Solving Equation (19)**
>
> The JKO scheme proves that the fixed-step Euler approximation of a continuous flow is equivalent to solving a sequence of locally constrained minimization problems in probability densities using the Wasserstein metric, thereby circumventing the inherent gradient instability associated with recursion in numerical ODE solvers.
> This insight is critical because it formally establishes that the update of the flow policy is a Wasserstein Gradient Flow, giving the update rule a theoretical foundation that guarantees a stable, trust-region-like policy improvement.
>
> **Solution of Equation (19):**
> The objective in Equation (19) is solved using standard sampling and optimization techniques over the neural network parameters $v_{\theta}$. The key is that the objective is designed to be locally tractable.
> * Sampling: We sample $(s, \mathbf{a})$ from the replay buffer and a discrete time step $t$.
> * Tractability: To minimization of Equation (19), we use **particle**-based solver. The detail is added to Appendix E.
> The $\mathbf{L}_2$ norm approximation is used for the $W_2^2$ term (as standard in Rectified Flow literature) to penalize the change in the velocity field, making the entire objective a simple $\mathbf{L}_2$ regression loss combined.
>
> ### **W3/W4/Q3 Locomotion Task and Comparison with Flow Q-Learning**
> We fully agree with the reviewer that including locomotion benchmarks and a direct comparison to Flow Q-Learning (FQL) is essential to confirm the robustness and practical relevance of our method. We have taken immediate action to strengthen the manuscript.
>
> To address the need for evaluation on diverse dynamics, we incorporate extensive evaluations on 5 challenging tasks from the OGBench suite, which includes a variety of complex locomotion control problems. The results are added in Appendix F.2. The new findings clearly demonstrate that SWFP achieves competitive or superior performance compared to FQL in these benchmarks. This robustly validates our design choice of using the JKO scheme for direct online policy refinement, positioning it as a highly effective and stable alternative to FQL’s decoupled training strategy across diverse robotic modalities.
>
> ### **Response to research integrity issues**
>
> We appreciate the reviewer for bringing the existence of Submission 691 to our attention and flagging the potential concern regarding submission overlap.
>
> **We confirm that we have no shared authorship with the authors of Submission 691, nor do we have prior knowledge of their work.** While both Flow Matching and JKO scheme for policy learning is overlap, our paper distinguishes itself from Submission 691 in several critical motivation and methodological aspects.
> * **Core Problem & Output Structure:** Our primary motivation is to solve the gradient instability issue inherent in pushing RL gradients through the iterative Flow Matching policy during online fine-tuning. Consequently, the output model of SWFP does **not change its structure or inference procedure** post-training (only the inference step count is fixed). In contrast, Submission 691 appears to introduce additional computation or complexity during the pushforwarding (inference) stage to model the value gradient flow.
> * **Computational Efficiency & Parallelism:** From a technical perspective, our method is designed to leverage the step-wise nature of the flow for parallelism. By defining the update locally, SWFP is able to achieve parallel training across flow blocks/time steps. Submission 691 appears to necessitate completing the sampling of all steps, potentially limiting its parallel efficiency.
>
> These differences in motivation and core implementation justify the independent nature of our contribution.

---

> > ### Comment · Reviewer_sK8u · 2025-11-27
> >
> > Thank you for the clarification. I now have a better understanding of the proposed algorithm.
> >
> > I appreciate the comparison with FQL and the evaluation on the locomotion task. However, could you also include FQL in the experiments presented in the main manuscript? The advantage of the proposed method over FQL seems marginal in the locomotion task.
> >
> > Regarding the relation to Submission 691, thank you for clarifying. Since there is no overlap among the authors, I have no further concerns on this point.

---

> > > ### Comment · Program_Chairs · 2025-11-27
> > > **Confirmed No Author Overlap**
> > >
> > > Confirming that there is no overlap between the authors on the two submissions.

---

> > > ### Author Response · Authors · 2025-11-28
> > >
> > > Thank you for your prompt response and for confirming your better understanding of our proposed algorithm.
> > >
> > > We appreciate your suggestion to include FQL in the core evaluation. We agree that this comparison is highly relevant as FQL is a contemporary flow-based RL method. To address this, we have supplemented **Figure 4** with the success rate curves for FQL during the offline-to-online training phase in the main manuscript, and the detailed hyper-parameters are listed in Appendix C.2.
> > >
> > > As the updated Figure 4 clearly illustrates, SWFP maintains a distinct advantage in convergence speed and final stability compared to FQL during the online fine-tuning phase, thereby contextualizing and highlighting our method's superior overall robustness and efficiency across diverse robotic challenges.
> > >
> > > Thank you once again for your constructive feedback.

---

### Official Review · Reviewer_3kYB · 2025-10-31

**Soundness:** 4
**Presentation:** 3
**Contribution:** 4
**Rating:** 8
**Confidence:** 3

**Summary:**

This paper proposes to learn flow policies using RL. Unlike previous work in behaviour cloning that learns flow fields that match the demonstration trajectories, this paper allows the flow field to be optimised against an explicit objective function. It does so tractably by using the JKO time discretisation scheme to generate incremental improvements to the flow field to match a maximum-entropy energy field encoding the reward function.

**Strengths:**

Overall this paper proposes an interesting and novel approach to learning flow fields. The paper is mostly clearly written (with occasional typos), and the paper is careful to take the reader from the original problem statement (above equation 3) through to the policy improvement step in equation 15.

The experimental results show that the technique performs well on several baselines.

**Weaknesses:**

The primary concern I have is how practical is the technique. I believe the authors when they say that recursive gradient computations are not tractable, but it is not clear to me how tractable each update of equation 15 is. It also the case that the learning process involves learning a soft critic for the target along side the policy -- the learned Q function is presumably updated after each JKO optimisation update. The Q function in SAC is usually updated using TD learning -- is that straight forward given this parameterisation?

A second concern I have is the novelty of the technique. Flow-based RL is not exactly novel, and there have been previous results such as Reinflow (Zhang et al 2025) and Flow q-learning (Park et al 2025). These are very recent, so I don't view this as a huge problem but it would be good to provide some kind of comparison in the response.

Given that the technique is fundamentally distribution matching between the free-energy functional of equation 3 and the Q functional of the policy, it is not clear if there are constraints on the reward function. Are there kinds of problems where this technique will not succeed?

The authors describe in the introduction that they view RL as a fine-tuning step on top of behaviour cloning, but many RL problems do not have an initial human-provided solution and the "fine-tuning" must actually be performed from scratch. It is not clear how strong the gradients are when pushed through the distribution matching.

It is interesting that the SWFP technique does not uniquely dominate all techniques in figure 4 --- it achieves comparable performance to the best technique in every domain and in some domains is the clear best performer (e.g., CALVIN, kitchen-complete-v0) but is not better than DPPO in Can and not better than IDQL for kitchen-partial-v0 --- the authors do not in fact describe figure 4, and so provide no insight into where SWFP is in fact a superior technique.

**Questions:**

Is the Q function learning for the soft critic updated after each JKO update? Do we have to do anything different to recover a TD step?
Are there kinds of problems where this technique will not succeed?
Given that SWFP does not outperform other techniques in some domains, what are the characteristics of problems where SWFP is superior?

---

> ### Author Response · Authors · 2025-11-21
>
> Dear Reviewer,
>
> Thank you for your detailed and thoughtful analysis of our work.
>
>
> ## **Weaknesses**
>
> ### **W1: Practicality and Tractability of JKO Update (Equation 15) /Q1/Q2**
>
> The core of the SWFP update (Eq. 15) is derived from the variational JKO scheme, which transforms the complex policy improvement problem into a local optimization problem. This local objective is composed of:
>
> * Minimize the KL divergence between the current step policy and the target energy-based policy.
> * The Wasserstein distance penalty ($W_2^2$), which is tractably approximated via a simple quadratic penalty on the difference between the current flow model's velocity field ($v_{\theta}$) and the proposed new velocity field ($v'$).
>
> Crucially, the policy update only requires gradient computation over the policy network's output at the sampled time step $t$, rather than propagating gradients backward through the full $N$ steps of the ODE solver. We give more detail in Appendix E.
>
>
> The reviewer is correct that the learning process involves a soft critic (Q-function). We confirm that the Q-function update is standard and decoupled from the policy optimization.
> The critic (Q-function $Q_{\phi}$) is updated using standard Bellman error without additional design (specifically, a soft Q-learning objective similar to SAC), as shown in eq.16-eq.18.
>
>
> ### **W2: Novelty and Comparison to Related Work**
>
> We acknowledge the growing field of Flow-based RL. Our core novelty lies in introducing the **Jordan-Kinderlehrer-Otto (JKO) scheme** to provide a principled, regularized, and scalable framework for **online policy refinement** of generative policies.
>
> * **Distinction from Reinflow (Zhang et al 2025):** In contrast to methods like ReinFlow, which convert the deterministic flow path into a stochastic process (SDE) by injecting learnable noise to enable standard policy gradient optimization, our Stepwise Flow Policy (SWFP) achieves online refinement by utilizing the inherent alignment between the fixed-step Euler discretization and the JKO scheme, providing a theoretically grounded framework for policy update in the Wasserstein trust region.
> * **Distinction from Flow Q-learning (Park et al 2025):** Unlike Flow Q-Learning (FQL), which addresses offline reinforcement learning by training **an auxiliary one-step policy** to guide the flow policy via distillation, our Stepwise Flow Policy (SWFP) focuses on online refinement by **straightforwardly** optimizing the parameters of the flow matching policy using a stable, theoretically grounded update mechanism in the Wasserstein probability space.
> * **Action:** We have updated the Related Work section to explicitly compare SWFP with these recent flow-based RL methods.
>
> ### **W3: Reward Constraints and Failure Modes / Q3**
>
> * **Reward Constraints:** Our method is built on the robust foundation of **Max-Entropy RL**. There are **no constraints** on the reward function beyond those typically assumed in standard RL (e.g., bounded rewards).
> * **Failure Modes:** Like any Q-learning method, SWFP's performance is susceptible to: 1) **Poor Q-function Estimation** (in environments with extremely sparse or complex reward functions). 2) **Poor Exploration** in the initial stages if the initial policy is weak and fails to find positive rewards.
>
> ### **W4: Starting from Scratch and Gradiant Strength**
>
> Our initial focus on fine-tuning aligns with the current paradigm in complex robotics and generalist models, where pre-training via IL/BC is a prerequisite for achieving useful performance, making fine-tuning a more practical and demanding technical challenge to solve robustly.
>
> However, the SWFP objective is flexible: the BC(flow matching) loss term and the RL objective can be simultaneously optimized from scratch, effectively supporting a variety of RL algorithms, including those used in offline RL settings. We have included offline RL results in Appendix F.1, where SWFP still achieves competitive results compared to established offline methods like FQL, IFQL, and IDQL.
>
> ### **W5: Performance Analysis and Superiority (Figure 4 / Q4)**
>
> We apologize for the oversight in describing Figure 4 in the main text. We have included a detailed discussion of these findings for Figure 4 in the Results section.
>
> SWFP's primary advantage is its **efficiency and consistency**. It achieves comparable or superior performance across all benchmarks while using **significantly fewer inference steps** ($N=5$ vs. $N \ge 50$ for diffusion-based methods), providing a massive advantage in **Clock-Time Efficiency** and scalability. In tasks like kitchen-partial-v0 and Can, SWFP initially lags (due to the reduced inference steps of the pretrained flow policy) but ultimately achieves a competitive final success rate comparable to or exceeding the best-performing baselines (DPPO and IDQL).

---

### Official Review · Reviewer_oc5Y · 2025-11-09

**Soundness:** 3
**Presentation:** 2
**Contribution:** 2
**Rating:** 4
**Confidence:** 3

**Summary:**

This paper formulates reinforcement learning as an iterative refinement process grounded in optimal transport and flow-matching theory. It interprets policy optimization as a Wasserstein gradient flow over probability distributions, where each update step minimizes a composite objective consisting of an energy functional (negative expected return with entropy regularization) and a transport regularization term. The authors derive a practical algorithm, termed Stepwise Flow Policy (SWFP), that discretizes this gradient flow into tractable updates and implements it using conditional flow-matching parameterizations. The method represents policies as learnable stochastic transport maps and performs block-wise optimization akin to proximal iterations in the space of distributions. The paper connects this formulation to existing soft policy iteration and diffusion-policy approaches, provides theoretical grounding through the Jordan–Kinderlehrer–Otto (JKO) scheme, and demonstrates implementation details for both policy and value updates within standard continuous-control benchmarks.

**Strengths:**

I am not an expert in the generative-policy literature, so my comments are based primarily on the content of this submission itself.

- The motivation of the paper is clearly articulated: in online policy iteration, it is generally infeasible to back-propagate through a large number of inference steps of flow-based policies. The authors address this limitation by proposing an alternative formulation based on Wasserstein gradient flow. (However, I have some reservations about this aspect, as noted in the weaknesses section below.)

- The introduction of a parallel training scheme that leverages replay-buffer data could be a significant contribution to online policy improvement for generative policies, assuming that prior work has not already addressed this direction.

- The derivation from Wasserstein gradient-flow theory to the practical implementation seems technically sound, and I did not observe any hand-wavy reasoning or questionable design choices.

**Weaknesses:**

- Empirical evaluation: The empirical evaluation appears somewhat unfair. While SWFP is tuned over a grid of hyper-parameters, the baselines are not subjected to comparable hyper-parameter search, as stated in the appendix. This imbalance in tuning effort makes it difficult to attribute the reported performance gains solely to the proposed method.

- On the motivation and possible baseline: Given that the policy network used in experiments is a relatively small MLP and the SWFP block size is only five, it should generally be feasible to back-propagate through the entire recursive sampling process. This makes the stated motivation, avoiding back-propagation through many inference steps, feel somewhat less compelling in the current experimental scale. Nonetheless, the topic remains relevant and important, especially considering the scaling trend in robotic foundation policies. To strengthen the paper, the authors could consider adding a baseline that directly back-propagates through the sampling process, which might serve as an approximate upper bound for online iterative refinement of generative policies.

**Questions:**

- L135 ``wherein $u_t(x\mid x1)$ becomes tractable by defining explicit conditional probability paths from $x_0$ to $x_1$, such as OT-paths or linear interpolation paths'': Could the authors clarify which was actually used in the implementation?


- L246 ``To solve the above problem, we first model the continuous-time diffusion process by a partial differential equations (PDE), i.e., the Fokker-Planck equation, $\partial \rho(t, x) / \partial t = \nabla (\nabla U(x) \rho(t, x)) + \beta \nabla^2 \rho(t, x)$'': I’m curious about the motivation for introducing the Fokker–Planck equation here. Is this formulation commonly used in generative-policy models, or is it mainly used as a convenient connection to the JKO scheme? Would it be helpful to provide further clarification or justification for this modeling choice in the paper?

- Figure 3: If I understand correctly, the target reward landscape consists of eight Gaussian components, each representing a high-reward region. In the right subfigure, however, the particles appear to concentrate around three modes (with one dominant and two minor clusters). Does this suggest that SWFP may not fully capture multimodal distributions? Ideally, one might expect convergence toward all eight modes. Could the authors clarify whether this behavior is expected or a limitation of SWFP. If it's incorrect, could the authors design another example showing how multi-model behavior can be well-captured?

---

> ### Author Response · Authors · 2025-11-21
>
> Dear Reviewer,
>
> Thank you for your thorough review and valuable feedback on our paper. We have carefully studied all your comments and questions, and we agree that they highlight potential ambiguities or limitations in the current manuscript. We believe the following clarifications and commitments to revisions will further strengthen our work.
>
> ## **Weaknesses**
>
> ### **W1: Empirical evaluation (Hyper-parameter tuning fairness)**
>
> We understand your concern regarding the lack of comparable hyper-parameter search for the baselines.
>
> * **Clarification and Rationale:** The hyper-parameters for baselines were **strictly adhered to their established settings** in original papers or authoritative benchmarks.
> * **Focus of Our Method:** **SWFP itself was not subjected to extensive tuning** beyond the Flow Block Size ($N$). As demonstrated in our Ablation Study, the method proves robust across a range of $N$ values (excluding the excessively small cases), highlighting the inherent stability of our Flow-based approach. The small optimal $N$ (e.g., $N=5$) used in our experiments is a key advantage, inherently demonstrating a significant clock-time efficiency over iterative generative methods like Diffusion Policies, which typically require dozens of diffusion steps.
> * **Commitment to Improvement:** We detail the sources and specific values of all hyper-parameters used for the baseline methods in the appendix.
>
> ### **W2: Motivation and possible baseline**
>
> Your point regarding the feasibility of direct back-propagation at the current small scale is well-taken.
>
> * Achieving strong performance with only $N=5$ Flow Steps is an advantage of SWFP, significantly shortening inference time. A naive flow matching policy would show the substantial performance degradation at such a low step count.
> * Direct Back-propagation Baseline: We agree to strengthen the paper by adding a direct back-propagation baseline. The experimental findings for the Direct Back-prop baseline (FBRAC) have been included in the appendix F.2. FBRAC use a linear combination of flow matching loss and the Q loss with backpropagation through time. The results on OGBench empirically demonstrate its poor performance in the online RL setting.
>
> ## **Question**
>
> ### Q1: L135 clarification: Conditional Flow Matching path
>
> We used the most common Linear Interpolation Path. The path is defined as $\mathbf{x}_{t} = t\mathbf{x}_{1} + (1-t)\mathbf{x}_{0}$, and the corresponding conditional vector field $\mathbf{u}_{t}(\mathbf{x}|\mathbf{x}_{1})$ is straightforward and computationally efficient. We clarify this in the revised version.
>
>
> ### Q2: L246 clarification (Fokker-Planck equation motivation)
> Your observation is very accurate. The motivation for introducing the Fokker-Planck (FP) equation is to establish a strict theoretical link between our method and the JKO scheme.
>
>
> We introduce the FP equation to: 1. Provide a continuous-time ODE that describes the evolution of our policy distribution. 2. Leverage the JKO scheme to discretize the policy optimization into a series of stable update steps that are regularized by the Wasserstein distance. We clarify this in the revised version.
>
> The density evolution through the CE(eq.2) and the FPE(eq.11) are mathematically equivalent when we set $v(x,t) = -\nabla U(x) - \nabla \log \rho(x,t).$ The core finding of the JKO scheme is that the Implicit Euler discretization of the FP equation is equivalent to solving a Proximal Operator problem in the Wasserstein space of probability measures.
>
>
>
> ### Q3: Figure 3
> We apologize for the confusion that Figure 3 has caused the reviewers. This necessitates a clear distinction between the pre-training objective and the online fine-tuning objective.
>
> **Clarification of Objectives:**
>
> * Figure 3 (left) shows the target distribution for the pre-trained Flow Matching policy, which is an 8-component Gaussian Mixture.
>
> * The role of SWFP is to online fine-tune this policy to shift samples towards high Q-value regions (high-reward actions). As shown in Figure 3 (right), the target Q-value landscape in this example was concentrated on only one mode (the dominant purple region in the top-right). SWFP successfully converged the policy distribution to this single high-Q region, confirming it is working as intended for efficient policy improvement, rather than failing to capture all eight modes.
>
> **Retention of Multimodality:** SWFP does not lose its ability to capture multimodal distributions. The final policy output remains a fixed-step Flow Matching model, which inherently retains this capability.
>
> **Commitment to Improvement:** We have included a new experiment in the Figure 3 using a multimodal reward. This result clearly demonstrates that SWFP can effectively converge the policy distribution to all high-Q modes, proving that our framework retains the intrinsic multimodal fitting capability of generative models during online policy fine-tuning.

---

### Meta-Review · Area_Chair_UHvc · 2025-12-22

**Summary:**

The main concern that stood out was the comparison with recent alternative flow-matching Q-learning algorithms, such as FQL (ICML 2025) [3kYB, sK8u]. The authors revised figures to compare with FQL, showing that the proposed method is roughly on par with FQL. The main benefit of the method highlighted in the rebuttals seemed was "efficiency and consistency". However, the actual experiments included seem to primarily measure returns, not "efficiency" or "consistency" or "clock-time efficiency" (i.e., metrics for these quantities don't appear in most tables, nor on the Y axis of most plots). The reviewers also raised concerns about hyperparameter tuning of the baselines and the novelty of the approach.

Given these concerns, I recommend that the paper be rejected. Overall, I think the connections between flow modeling and JKO are intriguing! In a future resubmission, I'd recommend that the authors focus on what specific benefits this new perspective brings, and to center experiments on highlighting those benefits. Note that this might mean that the figures do not look like the learning curves we're accustom to seeing in RL papers, but look more like targeted experiments aimed at answering specific research hypotheses.

**Reviewer Concerns:**

(see below)

**Reviewer Scores:**

oc5Y: 4 --> 4
* [-] no hyperparameter tuning of baselines: Authors clarified that the baselines were used in the same environment as the original papers (which presumably did hyperparameter tuning). However, this seems not to be entirely accuracy: I checked this by looking at Table 1, picked a baseline (Cal-QL), and found that the original paper did not compare to Robomimic (one of the tasks in this current paper). [The results on Kitchen Franka also seem to be different, but maybe this is because of the "normalization" mentioned in Table 1 of the CalQL paper]
* [+] "adding a baseline that directly back-propagates through the sampling process": The authors appear to implement this baseline and find it works quite a bit worse (Table 3 in the appendix).

3kYB: 8 --> 6
* [+] tractability of the proposed algorithm: authors clarified that the "policy update only requires gradient computation over the policy network's output at the sampled time step, rather than propagating gradients backward through the full N steps of the ODE solver. "
* [-] novelty relative to recent work (e.g., Reinflow, FQL): Rebuttal discussed conceptual algorithmic differences, and authors revised the related work section to discuss. However, Appendix Table 3 seems to indicate that the proposed method is not better than FQL (and is missing error bars, so I can't determine statistical significance). FQL was published at ICML 2025, 2 -- 3 months before the ICLR deadline.
* [/] Are there kinds of problems where this technique will not succeed? Authors argued that the only limitations are those inherited from Q-learning methods. I don't find this particularly compelling: presumably there are some other limitations (e.g., convergence of the ODE solver).
* [/] Is the method applicable to settings without demonstrations? Authored responded to this point by showing that the two losses (BC + Q maximization) can be optimized jointly, starting from scratch. I'm unsure if this addresses the reviewer's question, or if the reviewer's question was whether we can use just the Q-maximization term along (without the BC term).
* [-] Unclear if Fig 4 shows that the method is stronger than prior work (esp. DPPO): Authors clarified that the benefits are primarily computational savings, not higher asymptotic performance. However, I don't think that these claims are well substantiated in the paper. For example, if we look at the figures/tables in the paper, most report returns, not some metric of stability or computational gains. While the learning curves in Fig 4 give some sense of learning speed, they also indicate that one baseline DPPO learns _faster_ than the proposed method on about 4 / 6 tasks.

sK8u: 2 --> 2
* [-] "The presentation lacks clarity, and the paper is not well-organized.": I didn't see any discussion of this in the rebuttal
* [-] novelty relative to standard entropy-regularized RL: There was some discussion of this point, but I didn't see a concrete answer to the question of whether it's equivalent to entropy-regularized RL
* [/] missing algorithmic/experimental details: These were briefly discussed in the rebuttal.
* [-] missing comparison to FQL (including on locomotion tasks): The authors ran additional experiments comparing to FQL (Appendix Tables 3, 4), saying that "The new findings clearly demonstrate that SWFP achieves competitive or superior performance compared to FQL in these benchmarks." In Table 3, there are only 2/ 5 tasks where the proposed method achieves higher returns than FQL. The omission of error bars means the reader is unable to determine whether these results are statistically significant.

I am not taking into account the overlap with Submission 691, as the PC confirmed that that submission was independent.

---

### Decision · Program_Chairs · 2026-01-26

Reject